# Does Pancreatic Fistula Affect Long-Term Survival after Resection for Pancreatic Cancer? A Systematic Review and Meta-Analysis

**DOI:** 10.3390/cancers13225803

**Published:** 2021-11-19

**Authors:** Andrea Grego, Alberto Friziero, Simone Serafini, Amanda Belluzzi, Lucia Moletta, Luca Maria Saadeh, Cosimo Sperti

**Affiliations:** Department of Surgery, Oncology and Gastroenterology, 3rd Surgery Clinic, University of Padua, Via Giustiniani 2, 35128 Padua, Italy; d.andrea.grego@gmail.com (A.G.); albertofriziero@virgilio.it (A.F.); simone.serafini@ymail.com (S.S.); amandabelluzzi@gmail.com (A.B.); lucia.moletta@unipd.it (L.M.); lucamaria.saadeh@aopd.veneto.it (L.M.S.)

**Keywords:** postoperative pancreatic fistula, POPF, survival, overall survival, pancreatic cancer, pancreatectomy, pancreatic adenocarcinoma

## Abstract

**Simple Summary:**

The real influence of postoperative pancreatic fistula (POPF) on long-term survival after pancreatic cancer resection is unclear. The purpose of the present study was therefore to conduct a systematic review and meta-analysis of the impact of POPF on the disease-free and overall survival of patients with pancreatic cancer. Our results highlighted that clinically relevant POPF after surgery for PDAC seems to be significantly associated with shorter DFS and OS. Confirmation, with future studies, of a negative impact of POPF on survival may encourage the widespread use of risk-stratification tools for assessing fistula, centralization of patients, and probably a closer oncological follow-up.

**Abstract:**

Background: The impact of postoperative pancreatic fistula (POPF) on survival after resection for pancreatic ductal adenocarcinoma (PDAC) remains unclear. Methods: The MEDLINE, Scopus, Embase, Web of Science, and Cochrane Library databases were searched for studies reporting on survival in patients with and without POPF. A meta-analysis was performed to investigate the impact of POPF on disease-free survival (DFS) and overall survival (OS). Results: Sixteen retrospective cohort studies concerning a total of 5019 patients with an overall clinically relevant POPF (CR-POPF) rate of 12.63% (n = 634 patients) were considered. Five of eleven studies including DFS data reported higher recurrence rates in patients with POPF, and one study showed a higher recurrence rate in the peritoneal cavity. Six of sixteen studies reported worse OS rates in patients with POPF. Sufficient data for a meta-analysis were available in 11 studies for DFS, and in 16 studies for OS. The meta-analysis identified a shorter DFS in patients with CR-POPF (HR 1.59, *p* = 0.0025), and a worse OS in patients with POPF, CR-POPF (HR 1.15, *p* = 0.0043), grade-C POPF (HR 2.21, *p* = 0.0007), or CR-POPF after neoadjuvant therapy. Conclusions: CR-POPF after resection for PDAC is significantly associated with worse overall and disease-free survival.

## 1. Introduction

Postoperative pancreatic fistula (POPF) is the most common and feared complication following pancreatic surgery [1,2], with an incidence ranging from 5 to 20%. It was first defined in 2005 as the presence of amylase in the drainage fluid in quantities greater than three times the upper normal serum level, and cases were graded (A, B, C) by increasing clinical severity [3]. In 2016, the International Study Group for Pancreatic Fistula (ISGPF) refined this definition, omitting grade-A fistula inasmuch as it is the most favorable (healthy) clinical situation. POPF-related complications—such as intra-abdominal fluid collection, sepsis, and severe bleeding—may necessitate longer hospital stays, prolonged pharmacological treatment, and resurgery, and they carry the risk of organ dysfunction and even death [4]. Recent evidence suggests that POPF can play a role in the long-term survival of patients resected for pancreatic ductal adenocarcinoma (PDAC), affecting not only their short-term outcomes [5,6,7], but also their disease-free survival (DFS) and overall survival (OS). However, the real influence of POPF on long-term survival remains unclear. 

The purpose of the present study was therefore to conduct a systematic review and meta-analysis of the impact of POPF on the DFS and OS of patients with pancreatic cancer; to the best of our knowledge, this is the first meta-analysis analyzing this issue. The second aim of this paper was to represent a starting point for more systematic and less heterogeneous studies.

## 2. Materials and Methods

### 2.1. Search Strategy

A systematic literature search was conducted using the MEDLINE, Embase, Scopus, Web of Science, and Cochrane Library databases to identify all studies published up to 30 June 2021 regarding the impact of pancreatic fistula after surgery for PDAC. The search terms used were: “(pancreatic ductal adenocarcinoma OR pancreatic adenocarcinoma OR pancreatic cancer OR pancreatic carcinoma) AND (pancreatic fistula OR pancreatic fistula* OR pancreatic leak OR POPF) AND (survival OR long-term survival OR disease-free survival OR recurrence-free survival)” (Appendix A).

The titles and abstracts of all identified citations were screened by two investigators (A.F. and S.S.) to ascertain their relevance. Then, the full texts of potentially relevant articles were retrieved and separately examined by the same two authors (A.F. and S.S.). The reference lists in these publications were also reviewed (by authors A.G. and A.B.) to identify any additional articles suitable for inclusion.

The search method was developed from the Preferred Reporting Items for Systematic Reviews and Meta-Analyses (PRISMA) criteria [8].

### 2.2. Inclusion Criteria

To be included, studies had to meet the following criteria: (1) they reported on patients with a proven diagnosis of pancreatic adenocarcinoma, treated surgically with curative intent, with or without neoadjuvant chemotherapy and/or radiotherapy; (2) they considered POPF as defined by the ISGPF in 2005 or 2016 [3,4]; and (3) they examined the impact of POPF on OS and/or DFS.

The following were reasons for exclusion: (1) the absence of individual patient data; (2) reviews without original data; (3) a lack of long-term survival data; (4) short-term survival data, i.e., postoperative mortality (as a part of OS) before 30–60 days from surgery; (5) animal studies; (6) articles published in languages other than English.

### 2.3. Data Extraction

Two investigators (A.G. and L.M.) extracted the following data from each eligible study: study design, year of publication, study location and accrual period, total sample size, number of patients with a POPF according to the ISGPF criteria of 2005 or 2016, DFS (median) and OS (median) for the whole cohort and for patients with and without POPF, hazard ratios on univariate and multivariate analysis investigating DFS and OS in patients with and without POPF, patient demographics (e.g., sex, age), surgical procedure(s) performed (e.g., pancreaticoduodenectomy, distal pancreatectomy, or total pancreatectomy), R0 resection rates and presence of lymph node metastases, and proportion of patients given adjuvant and neoadjuvant treatment.

DFS was defined as the interval between the date of primary tumor resection and the date of recurrence, and OS as the interval between primary tumor resection and death or latest follow-up.

### 2.4. Risk of Bias Assessment

A methodological quality assessment of the studies considered was performed by two authors (A.G. and A.F.) using the Newcastle–Ottawa score (NOS) for assessing the quality of observational studies in meta-analyses [9]. This scale assigns a score of 0–9, based on the degree to which a sample is representative of the exposed cohort, and to which the cohorts are comparable, as well as controlling for confounding factors, the appropriateness of the outcome measures, selection bias, and reporting bias.

Discrepancies in judgments made were discussed between the two authors and, where consensus could not be reached, a third author was consulted.

### 2.5. Definition of Pancreatic Fistula

In the English literature, the ISGPF has defined POPF in two different ways—first in 2005, and then in 2016 [3,4]. According to the first consensus, POPF was diagnosed when the amylase content in the drainage fluid rose to more than 3 times the upper limit of normal in serum, starting from postoperative day 3. POPF was graded from A to C by increasing clinical severity. In the revised classification of 2016, grade-A POPF (conservatively treated biochemical leaks) was no longer considered, while grade-B and -C POPF were also defined as clinically relevant POPF (CR-POPF). For the purposes of the present paper, the term POPF is used as defined in the first ISGPF consensus, while CR-POPF is used to indicate grade-B or -C fistulas as stated in the later guidelines.

### 2.6. Statistical Analysis

Continuous variables were reported as means (SD) or medians (range), and categorical were reported as proportions. Adjusted hazard ratios (HRs) with corresponding 95% confidence intervals (CIs) were extracted from each study. Where no CIs were provided, they were calculated from the *p*-values [10]. If only median survival was reported, HRs and CIs were estimated [11]. A summary HR (95% CI) was calculated for patients with vs. without a POPF using a random-effects model estimated via the inverse variance method [12]. Heterogeneity between the studies was assessed using Cochran’s Q test (χ^2^), and was further quantified by generating an inconsistency statistic (I^2^) for each outcome measure. An I^2^ of 0–50% represented low heterogeneity, 51–75% represented medium heterogeneity, and 76–100% represented high heterogeneity. [13]. Sensitivity analyses were performed to explore potential sources of heterogeneity. For each outcome parameter with medium and high inter-study heterogeneity, individual studies were removed and the analysis was repeated to assess each study’s contribution to the overall effect size and heterogeneity. Funnel plot graphics were used to test for publication bias. Analyses were run on subgroups of patients in order to maximize population homogeneity. All analyses were performed with STATA 16/MP (Stata Corp. 2019. Stata Statistical Software: Release 16.0, College Station, TX, USA).

## 3. Results

### 3.1. Literature Search

The MEDLINE, Embase, Scopus, Web of Science, and Cochrane Library searches yielded a total of 4045 citations (Figure 1), and a consensus among all investigators led to 16 studies being judged as eligible for our meta-analysis [14,15,16,17,18,19,20,21,22,23,24,25,26,27,28,29]. Screening the references of the full-text articles assessed for eligibility revealed no further studies potentially suitable for inclusion. Eleven studies reporting DFS data (n = 3375 patients) [15,17,19,21,22,23,25,26,27,28,29] and sixteen studies containing information about OS (n = 5019 patients) were included in the meta-analysis [14,15,16,17,18,19,20,21,22,23,24,25,26,27,28,29] (Figure 1). Correlation between long-term survival and POPF was assessed as the primary endpoint in 14 studied [14,15,16,17,18,19,22,23,24,25,26,27,28,29]; only two studies considered all complications [21] and the analysis of prognostic factors [20] as primary endpoints, and POPF impact as a secondary endpoint.

### 3.2. Study and Patient Characteristics

Of the 16 retrospective cohort studies analyzed [14,15,16,17,18,19,20,21,22,23,24,25,26,27,28,29], 2 [19,29] were multicentric, and concerned a total of 5019 patients with an overall POPF rate of 20.78% (n = 1043 patients) and a CR-POPF rate of 12.63% (n = 634 patients). Table 1 and Appendix A show the main characteristics of these studies. Eleven of the sixteen studies reported patients’ ages (median or mean), which ranged from 61 to 71 years [15,17,18,19,21,22,23,24,25,26,29]. There was a male predominance in 12 of the 16 studies [14,15,16,17,18,19,20,21,22,23,25,26], with the proportion of males ranging from 51.6% [17] to 68.8% [22].

The surgical procedure most frequently used was pancreaticoduodenectomy, performed in 60–100% of patients in the various studies, followed by distal pancreatectomy and total pancreatectomy. Patients who underwent total pancreatectomy were not included in survival analysis in the two studies reporting these cases [21,23]. Four studies [14,17,22,25] only reported cases treated with pancreaticoduodenectomy, while Leon et al. only examined patients who underwent distal pancreatectomy [29]. R0 resection rates were reported in 10 of the 16 studies [17,18,19,20,21,22,23,24,25,26], and ranged between 57.4% [25] and 94% in patients pretreated with neoadjuvant therapy (NAT) [26]. Ten of the sixteen studies [17,18,19,20,21,22,23,24,25,26] reported on the presence of lymph node metastases, which ranged from 21% [20] to 74.9% [19].

### 3.3. Impact of POPF on DFS

Eleven studies [15,17,19,21,22,23,25,26,27,28,29] provided DFS data for patients with POPF or CR-POPF, as shown in Table 2 and Appendix A. The median DFS rates of whole cohorts, reported in only three studies [17,21,29], ranged from 12.2 [17] to 19 months [29]. Four studies described the impact of POPF (grades A–C) on DFS [17,22,23,25]: DFS was significantly shorter in the POPF group in the study by Dundar et al. [22], while no significant difference was seen in the other three studies [17,23,25]. Six studies reported the impact of CR-POPF (grades B and C) on DFS [15,21,25,27,28,29], with only Watanabe et al. and Nagai et al. reporting a significantly shorter DFS in their CR-POPF groups [15,21,28]. The one study that considered subgroups of patients with grade-C POPF [19] found no significant differences in DFS via multivariate analysis.

Nagai et al. [15] reported that the peritoneum was the predominant site of recurrence in patients with CR-POPF, while no such association was found by another four studies [17,18,25,29].

In the subgroup of patients given NAT, the only study that examined DFS found a higher risk of recurrence in patients with a CR-POPF [26].

Nine studies included in our meta-analysis analyzed the impact of POPF and CR-POPF on DFS [15,17,21,22,23,25,27,28,29]: four of them [17,22,23,25] assessed the impact of POPF on DFS survival, and six [15,21,25,27,28,29] examined the influence of CR-POPF. Two studies [19,26] were excluded from our meta-analysis of this issue because they only concerned patients with grade-C POPF [19] or POPF after NAT [26].

Our meta-analysis confirmed a significantly shorter DFS in patients with CR-POPF (summary adjusted HR 1.59, 95% CI 1.18–2.15, *p* = 0.0025), while for patients with POPF (grades A–C) the difference in DFS did not reach statistical significance (*p* = 0.12) (Figure 2). In most studies, hazard ratios were adjusted for tumor stage, grade, any presence of lymph node metastases or vascular invasion, R0 resection, and any administration of adjuvant therapy. Low statistical heterogeneity was noted in the pooled analysis, with an I^2^ test of 42%, as well as in the POPF and CR-POPF subgroups analyzed separately (I^2^ 53.5% and I^2^ 30.2%, respectively).

### 3.4. Impact of POPF on OS

Data regarding the influence of POPF or CR-POPF on OS were reported in 16 studies [14,15,16,17,18,19,20,21,22,23,24,25,26,27,28,29] (Table 2 and Appendix A). The median OS for all patients (with and without POPF) considered in the meta-analysis ranged from 19 months [14] to 25.4 [18] months, while for patients with POPF it ranged from 10.7 months [16] to 23.2 months [18]. There was only one study [20] in which OS was significantly lower for patients with a POPF than for the cohort as whole, irrespective of the grade of severity of the fistulas.

Seven of the sixteen studies analyzed OS in patients with CR-POPF, finding a median OS that ranged from 14.6 months [15] to 29.43 months [29]. OS was only statistically lower in patients with CR-POPF in two studies [21,27], while no significant impact of CR-POPF on OS came to light in the others. Three [19,20,27] studies analyzed survival separately for patients with grade-B POPF, grade-C POPF, or no fistula; in two of these studies [20,27], patients with grade-B POPF had a shorter OS than those with no complications (11 months vs. 21 [20]; 26 months vs. 28.6 [27]); in all three studies [19,20,27], patients with grade-C POPF had a significantly worse survival than the other patients. The hazard ratio for mortality among patients with grade-B or -C POPF ranged from 1.52 to 2.92.

Two studies [24,26] also considered the influence of CR-POPF on OS in patients given NAT; in this particular subgroup, patients with a CR-POPF seemed to have a mortality risk from 2.8 to 7.1 times higher than in patients without such fistulas.

The proportion of patients who had adjuvant therapy was reported in 11 of 16 studies [14,15,17,18,19,20,21,24,25,26,29], and ranged from 46.5% [24] to 85.5% [26] (Table 1 and Appendix A). Eleven studies [14,15,17,18,19,21,24,25,26,27,29] also examined the association between the onset of fistulas and the administration of adjuvant treatment. The presence of a POPF or CR-POPF was significantly associated with a lower rate of adjuvant treatment in four studies [19,21,26,27]. In particular, Kawai et al. and Bonaroti et al. found lower rates of adjuvant therapy, but only in patients with grade-C POPF [19,27]. Murakami et al. and Watanabe et al. reported that the time from surgery to adjuvant therapy was longer for patients with POPF, but this difference did not affect their survival [18,21].

Sufficient data for a meta-analysis were available in 16 studies [14,15,16,17,18,19,20,21,22,23,24,25,26,27,28,29] concerning 5019 patients. Eight studies [14,16,17,18,20,22,23,25] considered the impact of POPF on OS, and seven [15,21,24,25,27,28,29] examined the impact of CR-POPF, while three [19,20,27] focused on the effects of grade-B or -C POPF. NAT was only administered to a minority of patients, and was reported in six studies, two of which [24,26] examined the survival of patients with CR-POPF who were given NAT. These studies were thus analyzed separately. Medium statistical heterogeneity resulted in pooled analysis (I^2^ 64%). To reduce heterogeneity, a meta-analysis was performed on subgroups of patients based on the definition of pancreatic fistula adopted by the authors of the studies, resulting in a low statistical heterogeneity in all groups except for the grade-B POPF group, where high heterogeneity was present (I^2^ 80%). Among the various subgroups, our meta-analysis identified a statistically significant reduction in OS for patients with a POPF, with the exception of the subgroup with grade-B fistula (Figure 3). The hazard ratio for mortality ranged from 1.15 to 3.37 for patients with a POPF, and it was higher for those with grade-C POPF (summary adjusted HR 2.21; 95% CI 1.4–3.47; *p* = 0.0007), as well as for those previously given NAT and with CR-POPF (summary adjusted HR 3.37; 95% CI 1.63–7; *p* = 0.0011), (Figure 3). In most studies, the HR was adjusted for tumor stage, grade, presence of lymph node metastases or vascular invasion, R0 resection, and any administration of adjuvant therapy.

### 3.5. Risk of Bias Assessment

The scores on the Newcastle–Ottawa Scale (NOS) obtained by the studies included in our meta-analysis are summarized in Appendix A. While none of the studies were awarded the maximum NOS score (9 points), they all achieved acceptable scores of between 7 and 8. The quality of the studies was impaired mainly by their having considered scarcely comparable factors that might independently influence survival such as pre-existing comorbidities, which were not documented in all studies. When publication bias was assessed with funnel plots drawn for all subgroups meta-analyzed, no major asymmetries emerged—especially in the subgroups with at least 10 studies for which this test is appropriate—as shown in Appendix A.

### 3.6. Sensitivity Analysis

Sensitivity analyses were performed by removing individual studies from the data analysis of the two outcomes in order to assess for sources of heterogeneity. Excluding studies in this manner had no effect on the overall effects and heterogeneity of either outcome.

## 4. Discussion

An association between anastomotic leakage and a worse OS has been well established in patients with several types of tumor, including those involving the colon and rectum, the stomach, and the esophagus [30,31,32,33]. A recent meta-analysis by Mintziras et al. [34] also showed that severe postoperative complications in PDAC patients (and pancreatic fistula in particular) affect OS, increasing the likelihood of death by 45%. Nevertheless, POPF and its influence on the long-term survival of patients with PDAC remain a matter of debate.

The present study examined this issue. A meta-analysis of six studies (1441 patients) demonstrated that CR-POPF increased the risk of a recurrence of PDAC by 59%. CR-POPF—especially grade C—also negatively affects OS, as highlighted by a meta-analysis of nine studies (3545 patients), which found that patients with a grade-C POPF were twice as likely to die (summary adjusted HR 2.21; 95% CI 1.4–3.47; *p* = 0.0007) (Figure 3). This trend was also confirmed in patients given NAT [24,26].

The mechanism by which POPF affects survival in cancer patients remains unclear. The most likely reasons relate to delays, and the impossibility of delivering or completing adjuvant chemotherapy (ACT). Several authors [17,19,24,25] described adjuvant therapy as an independent risk factor influencing OS, so lower rates of adjuvant treatment could affect OS. That said, most authors specifically examining the relationship between POPF and survival found no association between the onset of a POPF and a lower likelihood of patients receiving ACT [14,15,17,18,24,25,29]. Only four studies [19,21,26,27] reported rates of ACT being lower among patients with CR-POPF (grade B or C). Be that as it may, some authors reported that starting ACT early after surgery improved OS [35,36,37], while others found that the time to the initiation of adjuvant treatment did not affect the survival of patients with PDAC [21,38,39]. The modification or discontinuation of oncological therapies could also significantly influence DFS and OS in this setting.

A recent large-scale study based on Medicare registry data identified a lower OS for patients with severe postoperative complications after complex cancer resection, even after adjusting for the use of adjuvant treatments [40]. This would suggest that other mechanisms influence oncological survival.

Postoperative sepsis seems to be associated with earlier cancer recurrence and, consequently, a lower OS [41,42]. Although the mechanisms behind this relationship have yet to be fully understood, the release of proinflammatory lymphocytes, cytokines, and chemokines secondary to a systemic inflammatory response syndrome (SIRS) induced by anastomotic leakage could well give rise to a state of immunosuppression, consequently allowing residual tumor cells to spread [43,44]. As seen in other cancers, the inflammatory response caused by anastomotic leakage (such as a POPF) may therefore increase the risk of local recurrences [31,32,45,46]. Nagai et al. identified CR-POPF as an independent risk factor for peritoneal recurrences of PDAC, and CR-POPF in these patients was associated with a lower OS [15]; however, other series did not confirm this impact of CR-POPF on the site of tumor recurrences [17,18,25,29].

Two studies [24,26] specifically distinguished between subgroups of patients who were or were not given NAT; an association between CR-POPF and a lower OS emerged only for the patients receiving NAT. In the present meta-analysis, we confirmed the major impact of CR-POPF on OS—especially in patients pretreated with NAT (summary adjusted HR 3.37; 95% CI 1.63–7; *p* = 0.001). Hank et al. [24] suggested that a POPF is less likely to occur in a pancreas with a fibrous texture. As NAT usually promotes pancreatic fibrosis, a CR-POPF may be associated with a lack of efficacy of NAT and, therefore, with a worse survival. Another possible explanation lies in an impaired immunological status following NAT being exacerbated by SIRS secondary to a CR-POPF, which would affect survival (as previously reported in gastroesophageal cancer). On the other hand, Uchida et al. [26] found that a subgroup of patients given NAT had a higher incidence of borderline resectable pancreatic cancer, a lower ACT rate, and a longer interval from surgery to chemotherapy. They concluded that the high rate of borderline resectable neoplasms, associated with ACT being administered after a delay or not at all, could explain the lower DFS and OS in patients with a CR-POPF. While all of these hypotheses are intriguing, they need to be validated by further studies on larger numbers of patients subgrouped by severity of POPF, margin resection, and NAT efficacy.

The negative impact of CR-POPF on the survival of patients who undergo surgery for PDAC might have important clinical implications, as suggested by the present meta-analysis. First, since CR-POPF (especially grade C) is an independent risk factor for shorter DFS and OS in patients with PDAC, major pancreatic resections should ideally only be performed at specialized centers by experienced pancreatic surgeons, in order to minimize the risk of this complication. Second, the worse prognosis associated with the occurrence of a CR-POPF could also be used as an argument for improving the development and application of fistula risk stratification (based on preoperative imaging, intraoperatively ascertained pancreatic texture, and intraoperative frozen section histology). A more widespread use of stratification tools could help surgeons to choose the most appropriate techniques to use in pancreatectomies (pancreatic duct stenting, pancreaticogastrostomy, reinforced staplers, etc.) in order to prevent grade-C POPF. Moreover, the correlation of POPF with lower DFS and OS could lead to a closer follow-up of these patients, so as to identify early relapse. Finally, these data could be used by surgeons in counselling patients as to the best surgical procedure(s), as well as for the assessment of surgical risk.

The present meta-analysis has some limitations that should be discussed. First, the retrospective nature of the studies considered could give rise to selection bias. Heterogeneity between the studies was largely due to the different accrual periods and consequent POPF definition. Moreover, studies with long follow-up times could be characterized by a modification of surgical techniques and oncological protocols that could influence patients’ long-term survival. Different surgical procedures were performed (pancreaticoduodenectomy and distal pancreatectomy), and they were frequently analyzed together despite different rates of POPF, clinical relevance, course, treatment and, finally, patient survival. The difficulty of interpretation and extrapolation of granular data such as cancer staging, adjuvant/neoadjuvant therapy administration and cycle completion, other postoperative complications, and patient comorbidities should also be acknowledged as a limitation of this study, and could be regarded as a source of bias. Therefore, we suggest that new studies should anonymously share data or report all of these features in order to perform subgroup analyses of patients with and without POPF.

## 5. Conclusions

In conclusion, CR-POPF after surgery for PDAC seems to be significantly associated with shorter DFS and OS. In particular, patients with grade-C POPF had twice the risk of dying compared with those without this complication. New prospective, multicenter studies are needed in order to clarify the impact of more or less severe POPF on DFS and OS after surgery for PDAC.

## Figures and Tables

**Figure 1 cancers-13-05803-f001:**
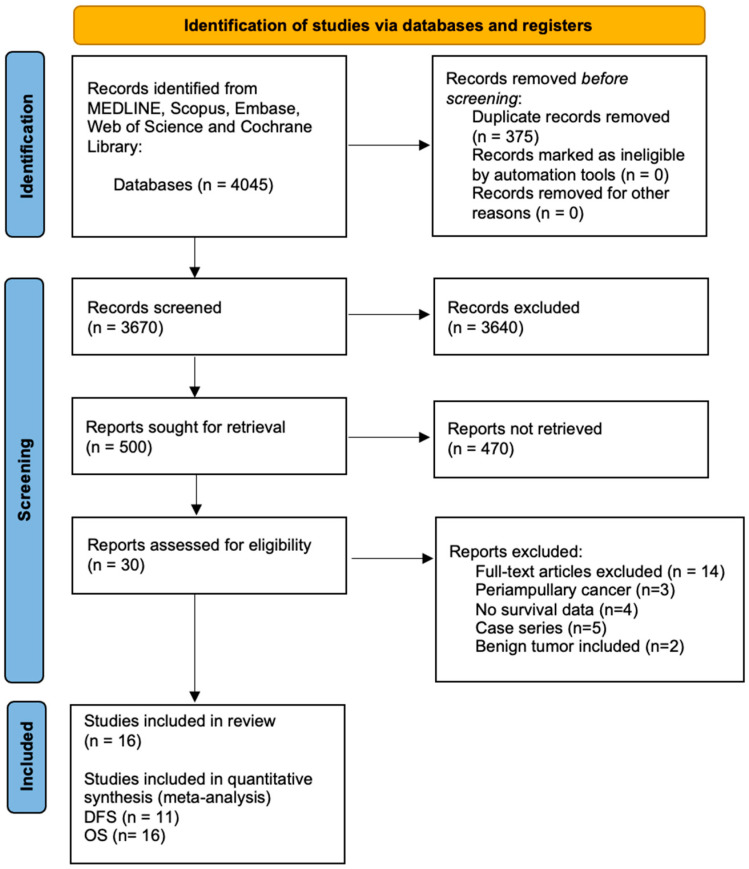
PRISMA flowchart showing the selection process of papers.

**Figure 2 cancers-13-05803-f002:**
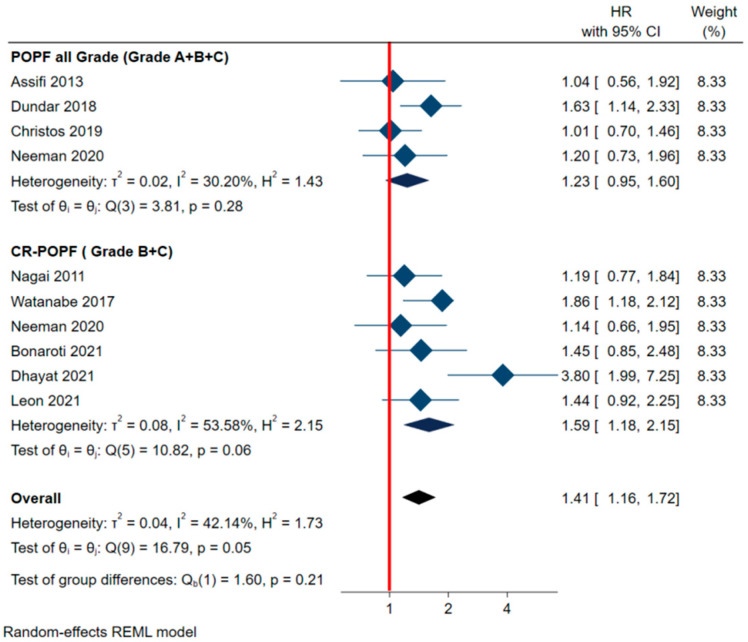
Forest plot summarizing the meta-analysis to compare disease-free survival in patients with and without postoperative pancreatic fistula (POPF) or clinically relevant postoperative pancreatic fistula (CR-POPF). HR: hazard ratio; 95% CI: 95% confidential interval.

**Figure 3 cancers-13-05803-f003:**
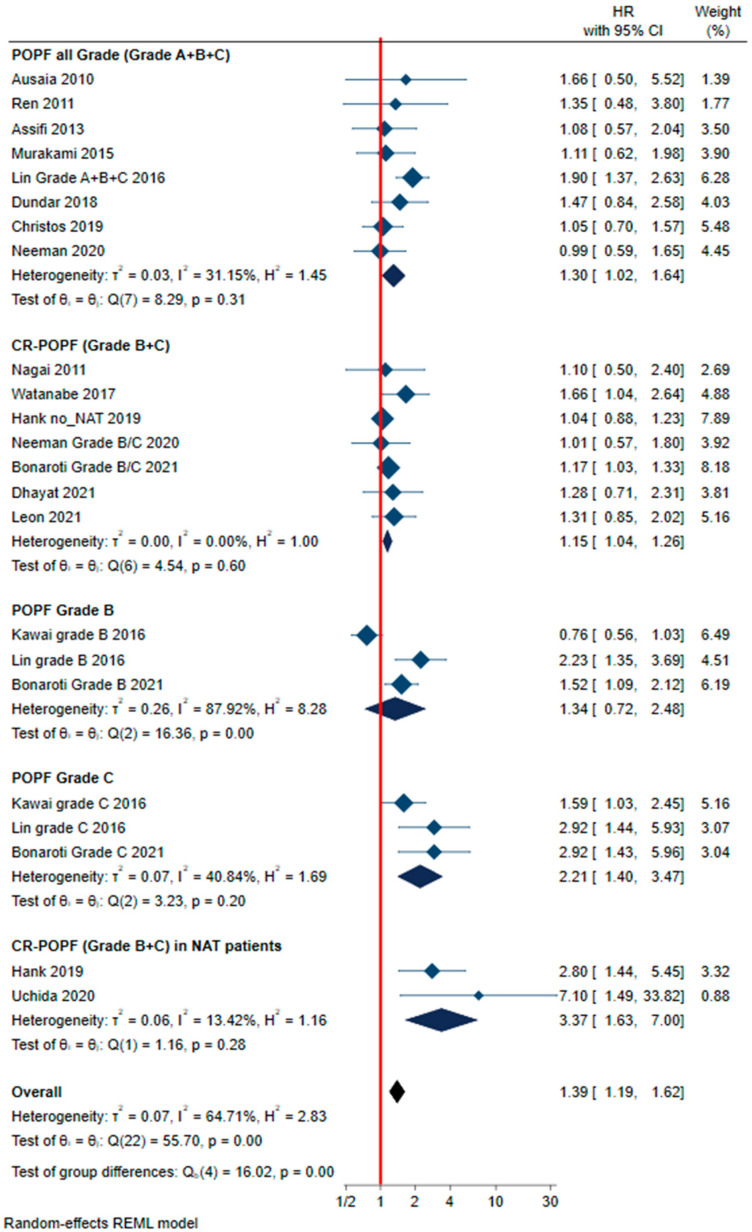
Forest plot summarizing the meta-analysis to compare overall survival in patients with and without postoperative pancreatic fistula (POPF), clinically relevant POPF (CR-POPF), grade-B POPF, grade-C POPF, or CR-POPF in patients who underwent neoadjuvant therapy. HR: hazard ratio; 95% CI: 95% confidential interval.

**Table 1 cancers-13-05803-t001:** General characteristics of the studies, the patients, and their tumors considered in the systematic review.

	NAT (n)	Patients	Surgery Type	All POPF	POPF B+C	ACT	NAT	LN+	R0
Author		n	n (%)	n (%)	n (%)	n (%)	n (%)	n (%)	n (%)
Ausania 2010	NO	47	PD, 47 (100)	9 (19.1)	7 (14.9)	26 (55.3)	0	NA	NA
[14]									
Nagai 2011	NO	184	PD, 152 (82.6)	51 (27.7)	46 (25)	128 (69.6)	0	NA	NA
[15]			DP, 32 (17.4)						
Ren 2011	NA	160	PD, 133 (83.2)	34 (21.3)	25 (15.6)	NA	NA	NA	NA
[16]			DP, 27 (16.8)						
Assifi 2013	NO	221	PD, 221 (100)	23 (10.4)	14 (6.3)	149 (67.4)	0	143	145 (65.6)
[17]								(64.7)	
Murakami 2015	NO	210	PD, 151 (71.9)	44 (20.95)	31 (14.76)	143 (68)	0	143 (68)	139 (66)
[18]			DP, 59 (28.1)						
Kawai 2016	YES+NO	1397	PD, 966 (69.2)	327 (23.4)	188 (13.5)	1122	359	896	1046 (74.9)
[19]			DP, 431(30.8)			(80.3)	(25.7)	(64.1)	
Lin 2016	NA	300	PD, 220 (73)	93	61	148	NA	64	240
[20]			DP, 69 (23)	(31)	(20.3)	(49)		(21)	(80)
Watanabe 2017			PD, 73 (60)	24	20	68	0	62	122
[21]	NO	122	DP, 47 (38)	(19.7)	(16)	(56)	0	(51)	(100)
Dundar 2018	NO	64	PD, 64 (100)	11	7	NA	NA	38	64
[22]				(17)	(10.94)			(59)	(100)
Christos 2019			PD, 195 (86.3)	69	28				
[23]	NA	226	DP, 23 (10.2)	(30.5)	(12.4)	NA	NA	113 (50)	188 (83.2)
Hank 2019	YES (346)	753	PD, 604 (80.2)	41 (11.85)	13 (3.8)	161 (46.5)	346 (45.9)	155 (44.8)	257 (74.3)
[24]	NO		DP, 149 (19.8)	134 (32.9)	28 (8.1)	283 (69.5)		305 (74.9)	239 (58.7)
Neeman 2020	YES+NO	148	PD, 148 (100)	29 (19.6)	17	107 (72.3)	6	77	85 (57.4)
[25]					(11.5)		(5.1)	(52.0)	
Uchida 2020	YES (52)	200	PD, 122(61)	NA	7 (13)	171 (85.5)	52 (23.6)	16 (31)	49 (94)
[26]	NO (148)		DP, 78 (39)		24 (16)			82 (55)	132 (89)
Bonaroti 2021	YES+NO	578	PD	NA	55 (9.5)	YES	YES	NA	NA
* [27]			DP		Grade C, 15				
Dhayat 2021	NO	126	PD	26 (20.6)	12 (9.5)	YES	0	NA	NA
* [28]			DP						
Leon 2021	YES+NO	283	DP, 283 (100)	42 (14.9)	51 (18)	234 (82.7)	42 (14.8)	NA	NA
[29]									

ACT: adjuvant chemotherapy; All POPF: postoperative pancreatic fistula grade A+B+C; DP: distal pancreatectomy; LN+: N1 status; m: months; NA: not available; NAT: neoadjuvant therapy; n: number of patients; PD: pancreaticoduodenectomy; POPF: postoperative pancreatic fistula; R0: no residual tumor; *: only data relating to pancreatic adenocarcinoma were extracted.

**Table 2 cancers-13-05803-t002:** Neoadjuvant therapy, modified adjuvant chemotherapy, disease-free survival, and overall survival in patients with and without postoperative pancreatic fistula.

	NAT	Influence of POPF on	Whole Population			Disease-Free Survival, m (95% CI)			Overall Survival, m (95% CI)		
Author		ACT?	OS m (95% CI)	DFS m (95% CI)	POPF Grade	POPF	No POPF	*p*-Value	POPF	No POPF	*p*-Value
Ausania 2010 [14]	NO	NO	19		All grades	88.9% *	65.8% *	0.244	16.5	27.5	0.411
Nagai 2011 [15]	NO	NO			Grade B+C	7.3	8.2	0.178	14.6	16	0.83
Ren 2011 [16]	NA	NA			All grades				10.7	17.1	0.312
Assifi 2013 [17]	NO	NO		12.2	All grades				17.8 (16–20.2)	17.2 (7.4–27.8)	0.52
Murakami 2015 [18]	NO	NO, but TTA was longer	25.4		All grades				23.2	25.7	0.743
Kawai 2016 [19]	YES+ NO	YES, in POPF C			Grade A		12.9			28.6	
		patients			Grade B	14		0.447	26		0.403
					Grade C	7.4		0.028	9		<0.001
Lin 2016 [20]	NA	NA	19		All grades				11	21	<0.0001
					Grade B				11		
					Grade C				1		
Watanabe 2017 [21]	NO	YES, in CR-POPF pts TTA was longer (not related with OS)	21	13	Grade B+C						
Dundar 2018 [22]	NO	NA			All grades	32.8 (16.5–49.1)	54.3 (45.8–62.8)	0.007	19 (10.5–27.5)	28 (19.9–36.2)	0.18
Cristhos 2019 [23]	NA	NA	23.13		All grades						
Hank 2019 [24]	YES	NA			Grade B+C				17	34	0.002
	NO	NO			Grade B+C				25	26	0.66
Neeman 2020 [25]	YES+ NO	NO	22		Grade B+C				22		0.96
Uchida 2020 [26]	YES	YES in NAT pts			Grade B+C						
	NO										
Bonaroti 2021 [27]	YES+ NO	YES			Grade B+C	10.37	15.03	0.175	21.3	24,93	0.02
Dhayat 2011 [28]	NO	NA			Grade B+C						
Leon 2021 [29]	YES+ NO	NO	35	19	Grade B+C	12	20	0.165	29.43	36.03	0.224
			(25.5–44.43)	(16.65–21.35)		(6.7–17.29)	(17.3–22.7)		(22.11–36.75)	(24.65–47.41)	

ACT: adjuvant chemotherapy; All grades: postoperative pancreatic fistula grades A+B+C; CR-POPF: patients with clinically relevant POPF (grade B+C); DFS: disease-free survival; m: month; NA: not available; NAT: neoadjuvant therapy; no POPF: patients with no postoperative pancreatic fistula; OS: overall survival; POPF: patients with postoperative pancreatic fistula; pts: patients; TTA: time to adjuvant therapy; 95% CI: 95% confidence interval; * = % of recurrences.

## Data Availability

No new data were created or analyzed in this study. Data sharing is not applicable to this article.

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
