# Peer review of "Does Pancreatic Fistula Affect Long-Term Survival after Resection for Pancreatic Cancer? A Systematic Review and Meta-Analysis"

_cancers, 2021, doi:10.3390/cancers13225803_

Round 1

Reviewer 1 Report

Very nice work !

There could be some biases by following issues:

Paragraph 92: patients with total pancreatectomy should be excluded. There is no reason for pancreatic fistula.

The same in Paragraph 156 and Table 1.

These patients with total pancreatectomy could lead to different results when analyzed in groups. Therefore they should be excluded from statistics.

Author Response

Dear reviewer. Thank you for your appreciation and for the interesting observation. Patients who underwent  total pancreatectomy were  included only in descriptive statistics, this category of patients are quite rare and were not included in survival analysis. To highlight this choice of the two studies reporting total pancreatectomy, we added the following chapter in section 3.2: “Patients who underwent total pancreatectomy, were not included in survival analysis in the two studies reporting these cases [21,23].  We deleted this data from table 1, without adding a new exclusion criteria.

Reviewer 2 Report

Thank you for the opportunity to revise this interesting paper. By the mean of a systematic literature review and metanalysis, Grego and co-workers analyse the impact of postoperative pancreatic fistula (POPF) in overall (OS) and disease-free survival (DFS) after pancreatic resection. Their findings are consistent with a detrimental impact of POPF on both OS and DFS.

Here are my comments:

  • From a methodological point of view, the review is correct, and Authors demonstrate a deep knowledge of the included papers. The manuscript is well written and logically organized and I found it very enjoyable
  • As also acknowledged by Authors, one limitation of this study is the inclusion of studies including both pancreaticoduodenectomies (PD) and distal pancreatectomies (DP), although the formers were more frequent. Despite most studies being adjusted for cancer staging, clinical relevance, course and treatment of POPF is very different between PD and DP. This makes authors findings less clear-cut. However, I acknowledge that Authors by necessity had to deal with available literature and that a subset analysis for any type of operation would not be feasible
  • Another word of caution is about the complex interplay between cancer biology/staging, postoperative complications, administration of neoadjuvant/adjuvant therapy, and the risk of recurrence. In my opinion, a review of this kind, lacking granular data, is still exposed to a risk of bias due to the included studies. For instance, the effect on DFS was mainly due to the inclusion of the article by Dhayat et al. (ref. 25) while the effect of OS, especially in patients with grade C POPF, could partially be due to postoperative mortality. This limitation, which appears to be mainly related to data availability, should be acknowledged
  • Despite being very interesting, I am not sure these findings would significantly impact on patient management. Pancreatic surgeons are well aware of the relevance of POPF and everyone tries to avoid it as much as possible. Instead, I think these data are important for patient counselling and surveillance.

Author Response

  • Dear reviewer, we appreciated your observation. Conduction of this meta-analysis was difficult because of all the reasons you highlighted and we have tried to reduce bias as much as possible, depending on with available literature. 
  • We agree that a limitation of our study was the inclusion of patients who underwent PD and DP. These two types of surgical procedure, as you mentioned, show different rates of POPF in terms of clinical relevance, course and treatment. Separate analysis is not possible from available literature. In our paper this limitation is only mentioned, for this reason we modify discussion as follow: “Different surgical procedures were performed (pancreaticoduodenectomy and distal pancreatectomy) and they were frequently analyzed together despite different rates of POPF, clinical relevance, course, treatment, and finally patient’s survival.”
  • Your observation about lacking granular data is correct and as you suggest we add this consideration in the discussion: “Difficulty of interpretation and extrapolation of granular data as cancer staging, adjuvant/neoadjuvant therapy administration and cycles completion, other postoperative complication and patients’ comorbidities should also be acknowledged as a limitation of this study and could be regarded as a reason of bias. Therefore, we suggest that new studies should anonymously share data or report all these features in order to perform subgroup analyses in patients with and without POPF". We agree with you about the effect of a single article on the DFS subset analysis. In particular, Dayat et al. report the greatest negative impact of POPF on DFS.  Watanabe at al. report also a statistically significant reduction of DFS in POPF patients. However, the other papers highlight a negative influence of POPF on DFS even though they do not reach the statistical significancy. Considering patients with grade C POPF, in this meta-analysis, postoperative mortality (30-60 days before surgery) was not considered in the computation of overall survival; in fact, all patients who died before at least 30 days from surgery were excluded from the studies. We modified exclusion criteria of paper to specify the general (long term survival of inclusion criteria) as follow: (4) short term survival, i.e. postoperative mortality (as a part of OS) before 30-60 days from surgery.
  • Finally, we agree with you that our results are important for patients counselling and surveillance, for example a closer follow-up in patients with a history of POPF. We modify discussion as follow: “Moreover, POPF correlation with a lower DFS and OS could lead to a closer follow-up of these patients, to identify early relapse. Finally, these data could be used by surgeon in counselling patient for best surgical procedure and for the assessment of surgical risk”. We think that this meta-analysis could encourage a centralization of patients with PDAC to obtain reduction of POPF rates and improvement of its treatment.   

Reviewer 3 Report

The question if a post-operative pancreatic fistula influences the oncologic outcome of the patients has a significant clinical relevance. The authors try to approach this via a meta-analysis.

The simple summary requires revision.

Introduction:

Lines #50 to #55 are already in the analysis mode and do not belong here. It should be clearly worded what are the differences of the here presented analysis to the already existing analyses (unique features).

Results:

Table 1 presents a lot of information but is designed unclear. Here, streamlining would be beneficial or the consideration to distribute the information over two tables; for example: sex ratio is not an evaluation point and could therefore be omitted. Given that all studies are RCS the column with this information could be taken out as well. The relevant number of POPFs should be presented more prominent and not shown in the last two columns of table 1. With this it will be more obvious that the number of the B and C POPFs is not even in the two-digit range in some studies.

The results should clearly present which of the evaluated studies have POPF as primary endpoint and which show this as secondary or tertiary endpoint. It should be discussed if the POPF as such or the resulting delays in start of adjuvant therapy represent a risk factor. Here the authors are speculating too much (line #302 to #310). Either this information can be seen in the analysis of the studies, or it should not be speculated about.

Especially in studies with many patients (ref. #11 and #14), evaluating >10 years it should be considered in the analysis if these patients remain comparable over this long timeframe or if essential operative aspects changed during this time?

The analysis as such is important. In the here presented form remain too many open questions that a publication in CANCERS cannot be supported.

Author Response

Dear reviewer, thank you for your interesting observations.

We modified simple summary

With regards to

  • Introduction: we deleted lines #50 to #55 that are just presented in Discussion. To our knowledge, this is the first systematic review and meta-analysis that investigate correlation between POPF and patients' long term survival.

  • Results: we appreciated your observation and we modified table 1 as you suggested, adding a new table to supplemental files. You correctly highlighted that probably not all studies were conducted with the clear intention of evaluating POPF impact on long term survival. In fact, only two studies (Watanabe et al and Lin et al.) did not take into consideration POPF impact as first end point in their studies.  All the others Authors (all in retrospective forms) declared it as the main object of their analysis. Retrospective cohort studies methodologically have not a primary or secondary endpoint in contrast to prospective studies/clinical trials. All data were collected, described and analyzed; given survival as the dependent variable; all the independent variables (for example POPF, adjuvant/neoadjuvant therapy, complication and others) were analyzed in univariate and multivariate model. However, we added this following text in the Results:” Correlation between long-term survival and POPF was assessed as primary end point in fourteen studied [14-19, 22-29], only two studies considered all complication [21] and the analysis of prognostic factors [20] as primary end point and POPF impact as secondary endpoint.”

  • From line #302 to #310 we reported consideration made in some papers included in meta-analysis. Your observation about speculation is correct and for this reason we deleted these lines from the text

  • With regards to the inclusion of studies with an evaluating times > 10 years, your observation is pertinent that is the reason why we added another possible limitation of the study: “Moreover, studies with long follow-up times, could be characterized by a modification of surgical techniques and oncological protocol that could influenced patients long term survival”.  As a matter of fact,  Kawai #11 paper described a cohort of patients from 7 high volume center, where two surgical techniques were described ( “Pancreatic anastomosis, a 2-layer or single-layer fashion, including pancreaticojejunostomy or pancreaticogastrostomy, and the use of pancreatic duct stents were selected according to each surgeon’s decision”.) and only an adjuvant protocol, when used, was present (“Postoperative adjuvant treatment using gemcitabine- or S-1-based chemotherapy was provided to patients”). Hank et al #14 reported a similar situation.High volume centers, and the use of similar surgical techniques, may correlate with a similar POPF rate; for this reason and in relation to a similar adjuvant chemotherapy protocol, patients operated during ten years period may be comparable in term of OS. In fact , none of the technical variations of pancreatic anastomosis, such as duct-mucosa, invagination method, and binding technique or pancreatic-gastric anastomosis, have found to be consistently superior  [Shailesh V. Shrikhande, Masillamany Sivasanker, Charles M. Vollme, et al. Pancreatic anastomosis after pancreatoduodenectomy: A position statement by the International Study Group of Pancreatic Surgery (ISGPS), Surgery (2017), 161 (5): 1221-1234, https://doi.org/10.1016/j.surg.2016.11.021.]
